# Contrast Medium Hypersensitivity: A Large Italian Study with Long-Term Follow-Up

**DOI:** 10.3390/biomedicines10040759

**Published:** 2022-03-24

**Authors:** Eleonora Nucera, Giuseppe Parrinello, Sebastiano Gangemi, Alessandro Buonomo, Arianna Aruanno, Franziska Michaela Lohmeyer, Riccardo Inchingolo, Angela Rizzi

**Affiliations:** 1UOSD Allergologia e Immunologia Clinica, Dipartimento Scienze Mediche e Chirurgiche, Fondazione Policlinico Universitario A. Gemelli IRCCS, 00168 Rome, Italy; eleonora.nucera@policlinicogemelli.it (E.N.); p.giuseppe@email.it (G.P.); alessandro.buonomo@policlinicogemelli.it (A.B.); arianna.aruanno@policlinicogemelli.it (A.A.); angela.rizzi@policlinicogemelli.it (A.R.); 2Medicina e Chirurgia Traslazionale, Università Cattolica del Sacro Cuore, 00168 Roma, Italy; 3Department of Clinical and Experimental Medicine, School and Operative Unit of Allergy and Clinical Immunology, University of Messina, 98125 Messina, Italy; sebastiano.gangemi@unime.it; 4Direzione Scientifica, Fondazione Policlinico Universitario A. Gemelli IRCCS, 00168 Rome, Italy; franziskamichaela.lohmeyer@policlinicogemelli.it; 5UOC Pneumologia, Dipartimento Scienze Mediche e Chirurgiche, Fondazione Policlinico Universitario A. Gemelli IRCCS, 00168 Rome, Italy

**Keywords:** contrast media, hypersensitivity, immediate hypersensitivity, delayed hypersensitivity, iodinated contrast media, gadolinium-based contrast media, comorbidity, skin tests, premedication, follow-up

## Abstract

Hypersensitivity reactions (HRs) to contrast media (CM) are a major problem. We compared differences of HRs to iodinated contrast media (ICM) versus gadolinium-based contrast media (GBCM), collecting data on prevalence, type, latency and severity. Secondly, the predisposition to perform new contrast tests, use of premedication and possible appearance of new reactions were explored in a long-term follow-up of 5 years. Clinical data, comorbidities, skin test (ST) results, re-exposure to CM procedures with any new reactions, premedication and CM used were collected. In a retrospective single-center study, 350 patients with mild to moderate HRs were enrolled. Asthma, food allergy, non-allergic drug hypersensitivity and neurologic disease were significantly more frequent in patients with HRs to GBCM compared to the high evidence of cardiovascular disease and history of cancer in patients with HRs to ICM. A marked delay in performing STs was reported by patients with negative results (66 months, *p* < 0.01). Iomeprol, iopamidol and gadobenic acid were the culprit CM most involved in HRs in patients with positive STs. During follow-up, 7.1% of responders reported new HRs to CM despite negative STs, premedication and infusion of alternative CM in most cases.

## 1. Introduction

The contrast medium (CM) is a drug, of various nature and composition, which is used in radiological diagnostic methods to increase the efficiency (i.e., the difference in contrast with respect to the surrounding area) of imaging a particular organ or body district [1].

In modern medicine, iodinated contrast media (ICM) and gadolinium-based contrast media (GBCM), are increasingly used for diagnosis and disease monitoring [1].

The iodinated contrast media include (1) non-water-soluble suspensions of diiodopyridine, (2) oily contrast agents and (3) water-soluble iodinated contrast agents. The latter, derivatives of triiodobenzoic acid, are subdivided into ionic, with high osmolality, and non-ionic iodates with low osmolality, nowadays the most used in medical practice. From a chemical point of view, ICM are classified into a monomeric structure if they have a benzene ring or a dimeric structure if the benzoic nucleus is covalently bound (Figure 1a).

Gadolinium-based contrast media use the paramagnetic properties of heavy metal element gadolinium to provide contrast in magnetic resonance imaging (MRI) studies. All GBCM consist of a gadolinium ion (Gd3+) complexed with a chelating ligand: a carrier molecule, the purpose of which is to remain bound to Gd3+ until it is excreted, preventing deposition of Gd3+ in tissues. Structurally, GBCM can be classified into linear versus macrocyclic and non-ionic versus ionic (Figure 1b).

More than 70 million doses of ICM and about 50 million doses of GBCM are administered worldwide per year [2]. Adverse reactions to CM are a relevant problem, which are in rare cases even fatal. Two types of adverse reactions to CM can be distinguished: (a) toxic reactions, which are predictable, dose-dependent and related to chemical properties of CM, and (b) hypersensitivity reactions (HRs), which are considered not predictable and dose-independent. HRs can be divided into “immediate reactions” if symptoms begin immediately up to 1 h after drug administration and into “non-immediate or delayed reactions” if symptoms start more than 1 h or up to 10 days after administration of the culprit agent. Immediate HRs (IHRs) have been reported in 0.7% to 3% of patients receiving non-ionic CM, severe reactions in 0.02% to 0.04% of intravenous procedures, and fatal IHRs in 0.00001% to 0.0003% of ICM administrations. The rate of HRs (mostly immediate) to GBCM ranges from 0.07% to 0.3%, whereas the rate of severe IHRs ranges from 0.003% to 0.008% with a death rate of less than 1 in a million [3]. The rate of non-immediate hypersensitivity reactions (NIHRs) ranges from 0.5% to 23% of ICM-exposed patients [2]. Regarding pathogenesis, IHRs to CM can be caused by IgE-mediated or non-IgE-mediated mechanisms.

Several different risk factors for IHRs and NIHRs to CM are being discussed, but they are not yet fully established. The most important and commonly agreed-on risk factor for HRs to CM is a previous reaction to CM (for both ICM and GBCM). Instead, the role of other risk factors, such as a history of atopy (e.g., asthma, food allergy, drug hypersensitivity), or other non-allergic comorbidities, such as cardiovascular diseases, tumors, autoimmune diseases or psychiatric disorders, remains controversial [2,3,4,5].

Clinical manifestations of IHRs to CM (both ICM and GBCM) include erythema and urticaria with or without angioedema, which occurs in more than 50–90% of patients. More severe symptoms include dyspnea, nausea, vomiting and hypotension [4,5]. In the most severe cases, anaphylactic shock and acute coronary syndrome (Kounis syndrome) are described [6]. The severity scales of Ring and Messmer or Brown can be used to classify these reactions [7,8]. NIHRs to ICM normally occur in the first 3 days after the administration of ICM; they are usually mild to moderate and generally resolve within 7 days. In these types of reactions, maculopapular exanthema (MPE) is the most common skin manifestation (30–90%), followed by delayed urticaria, with or without angioedema. Moreover, contact dermatitis; fixed drug eruption; and more severe manifestations, such as Stevens–Johnson syndrome/toxic epidermal necrolysis (TEN), acute generalized exanthematous pustulosis (AGEP), drug-related eosinophilia and systemic symptoms (DRESS), symmetrical drug-related intertriginous and flexural exanthema, graft-versus-host disease, vasculitis, neutrophilic dermatosis and iododerma, are reported [3,5,9]. NIHRs to GBCM are rarely reported; cases of maculopapular exanthema and AGEP to gadobutrol confirmed by biopsy and patch test were recently described [10,11].

The primary purpose of this large retrospective study was to compare patients with a history of HR to ICM with patients with a history of HR to GBCM. Patients underwent allergy testing to evaluate differences or similarities in their characteristics and prevalence, type, latency and severity of reactions.

Secondly, the study aimed to evaluate the predisposition to perform new contrast examinations, use of premedication and possible appearance of new reactions in a long-term follow-up of 5 years.

## 2. Materials and Methods

### 2.1. Patients

All patients who were referred from January 2016 to April 2021 to the Allergy Unit of the Fondazione Policlinico Universitario A. Gemelli IRCCS in Rome (Italy) with a compatible clinical history of HRs to CM (both ICM and GBCM) and who underwent an allergological evaluation were included in the study.

Clinical data were obtained from outpatient or medical records at the time of first evaluation according to the European Network of Drug Allergy (ENDA) questionnaire [12]. HRs to CM were classified as IHRs (occurring ≤ 1 h after CM administration) and NIHRs (occurring > 1 h to 7 days after CM exposure). The Ring and Messmer classification was used to classify the severity of IHRs [13]. Instead, NIHRs were classified according to Brockow’s proposal [14] as mild when no treatment was required, moderate when the patient responded readily to appropriate treatment and no hospitalization was needed, and severe when the reaction required hospitalization or was life-threatening.

The study was conducted according to the guidelines of the Declaration of Helsinki and approved by the Ethics Committee of the Fondazione Policlinico Universitario A. Gemelli IRCCS in Rome, Italy (ID 4212; Prot N.0027212/21). All patients gave written informed consent.

### 2.2. Comorbidities

All patients included in the study, who reported suggestive history of respiratory allergies or drug, food, Hymenoptera and contact adverse reactions, underwent a complete allergological evaluation including skin tests (STs) and measurement of specific serum IgE (UniCAP-Phadia, Thermo Fisher, Uppsala, Sweden) commercially available to diagnose other allergic diseases.

### 2.3. Methods

#### 2.3.1. Skin Tests

Allergy test data were obtained from STs performed with a set of up to a maximum of 13 different CM tested: 7 for ICM (iobitridol, iodixanol, ioversol, iopamidol, iopromide, iomeprol and iohexol) and 6 for GBCM (gadobutrol, gadobenic acid, gadoteridol, gadopentetic acid, gadoteric acid and gadoxetic acid).

Skin prick tests (SPTs) were performed with an undiluted commercially available solution and, in case of negativity, intradermal tests (IDTs) were performed. Evaluation for IHRs was performed 20 min after IDT at a 1:10 dilution, and for NIHRs delayed reading of SPTs or IDTs with 1:10 dilution, and if negative, IDT with undiluted CM at 48–72 h was performed.

Moreover, in the case of NIHRs, patch tests (PTs) with undiluted CM with reading at 48–72 h were performed in accordance with guidelines [2,5,15]. For immediate reading, SPTs were considered positive if, after 15 min, the size of the wheal was at least 3 mm in diameter with surrounding erythema. IDTs were considered positive when the size of the initial wheal after injection of 0.05 mL increased by at least 3 mm in diameter with surrounding erythema after 20 min [16]. Delayed reading of STs and PTs was performed according to international guidelines of the European Society of Contact Dermatitis [15].

#### 2.3.2. General Recommendations and Premedication

All patients evaluated were advised to avoid CM involved in the initial reaction(s), if known, and, in the alternative, to use a negative skin-tested CM in accordance with recent recommendations [2,5,17].

A premedication with an oral second-generation H1-antihistamine (e.g., cetirizine 10 mg) plus an oral corticosteroid (e.g., methylprednisolone 32 mg) once a day to be started 48 and 24 h before the examination and an intramuscular first-generation H1-antihistamine (i.e., chlorpheniramine maleate) plus a corticosteroid (i.e., methylprednisolone) 1 h and 30 min before the examination, respectively, was indicated for all patients.

### 2.4. Follow-Up

A phone survey was performed after at least 3 months to a maximum of 5 years from the allergological evaluation, during which patients were asked (1) whether they underwent a new exam with CM and (2) if there were new reactions to CM; if the first two questions were answered with yes, patients were asked (3) which CM was involved in any new reaction and (4) if they had taken premedication. Hospital medical records were evaluated for all patients for potential re-exposure to CM, and tolerance was verified and interpreted as negative if no adverse event was reported by the patient and medical records.

### 2.5. Statistical Analysis

Clinical, anthropometric and demographic characteristics were reported as means and standard deviations for continuous variables and as frequencies and percentages for categorical variables. Quantitative data were expressed as medians with range or interquartile range (IQR) with 25th and 75th percentiles (Q25–Q75 IQR, not normally distributed data assessed with a Shapiro–Wilk test). Discrete variables were compared using chi-square analysis or a Fisher exact test. Mann–Whitney U test was used for quantitative variables.

Statistical analyses were performed using the IBM SPSS software package, version 20 (SPSS Inc., Chicago, IL, USA). A *p*-value < 0.05 was considered significant.

## 3. Results

### 3.1. Patients

#### 3.1.1. Demographic and Clinical Characteristics

Three hundred fifty patients (255 females), with a mean age at the first evaluation of 57 years (±20, range 7–89), were included in the study. The median delay between onset of reaction and allergological evaluation was 62 months (range 3–72). Among the enrolled patients, 278 (79%) reported HRs to ICM, while 65 (19%) reported HRs to GBCM. Finally, seven patients (2%) described HRs to an unknown CM. Seventeen patients reported at least two HRs. Therefore, 368 HRs to CM were included in the analysis: 294 (80%) IHRs and 74 (20%) NIHRs. Most of the immediate and non-immediate reactions were mild to moderate (Table 1).

#### 3.1.2. Comorbidities

In the study population, the presence of both comorbid allergic/hypersensitivity and comorbid non-allergic diseases was explored. Non-allergic drug hypersensitivity, 113 (32%) patients, and allergic rhino-conjunctivitis, 65 (19%) patients, were the most frequently reported comorbid allergic/hypersensitivity conditions (Figure 2a). Regarding comorbid non-allergic diseases, a history of cancers and heart/cardiovascular diseases were the most frequently reported conditions (Figure 2b).

Table 2 shows the comparison between the group of patients with HRs to ICM and the group of patients with HRs to GBCM.

The two groups of patients differed in median age at the first evaluation (59 years, ±16, and 47 years, ±16, respectively; *p* < 0.01, Mann-Whitney U test).

No significant differences in type (immediate versus non-immediate), latency or severity of reactions (data not shown) were noted between the two groups.

Asthma, food allergy and non-allergic drug hypersensitivity were more frequent among patients with HRs to GBCM (14.5% vs. 5.3%, *p* < 0.05; 8.7% vs. 1.8%, *p* <0.01; 46.4% vs. 27.4%, *p* < 0.01, respectively; chi-square test). Regarding non-allergic comorbidities, cardiovascular disease and history of cancer were reported more frequently by patients with HRs to ICM (27.7% vs. 5.8%, *p* < 0.01; 56.6% vs. 40.6%, *p* < 0.05, respectively; chi-square test). Finally, patients with HRs to GBCM reported neurological diseases more often (23.2% vs. 11%, *p* < 0.05; chi-square test).

#### 3.1.3. Culprit Contrast Media

The retrospective analysis of the enrolled patients allowed identifying 289 HRs towards ICM and 72 HRs towards GBCM. A specific contrast agent, implicated in the initial reaction, was reported in 127 of 289 (43.9%) HRs to ICM and in 25 of 72 (34.7%) HRs to GBCM. The most frequently involved iodinated contrast agents were iomeprol (18%) and iopromide (17.3%), while gadoteridol (9.7%) followed by gadobenic acid and gadoteric acid (both 8.3%) were the most frequently reported gadolinium-based contrast media (Table 3).

#### 3.1.4. Skin Tests

Subsequently, the hypersensitivity reactions to CM were classified according to positive or negative outcomes of STs (Table 4).

Thirty of three hundred fifty enrolled patients had positive ST results. Regarding the comparison of patients with positive STs versus patients with negative STs, a significantly greater median delay in testing was noted in the group of patients with negative STs (66 vs. 22 months, *p* < 0.01, Mann–Whitney U test). Furthermore, in the group of patients with positive STs, a higher frequency of HRs to iomeprol was confirmed compared with patients with negative STs (45.4% vs. 15.7%, *p* < 0.01, Fisher exact test); an equally significant difference regarded iopamidol (13.6% vs. 1.5%, *p* < 0.05, Fisher exact test). Finally, the percentage of patients with negative STs for which it was not possible to identify a specific responsible ICM was significantly high (59.9% vs. 9.1%, *p* < 0.01, Fisher exact test).

Comparing patients with HRs to GBCM with positive versus negative STs, (1) a higher number of HRs in patients with negative STs and (2) a significantly higher percentage of patients with negative STs for which it was not possible to identify a specific responsible GBCM (70.3% vs. 25%, *p* < 0.01, Fisher exact test) were confirmed. Finally, the comparison among various GBCM reported identified gadobenic acid as more frequently involved in patients with positive STs (42.8% vs. 4.7%, *p* < 0.05, Fisher exact test).

#### 3.1.5. Follow-Up

The telephone survey, carried out after a minimum of 3 months to a maximum of 5 years from the allergological evaluation, allowed us to include 312 of 350 (89.1%) patients with HRs to CM into the analysis (Table 5).

Two hundred forty-six patients (79% of responders) reported an HR to ICM, while 61 patients reported an HR to GBCM (20% of responders). More than half of the responders (141 out of 246, 57%) with previous HR to ICM underwent a new contrast examination. Similarly, the percentage of responders with previous HR to GBCM (24 of 61 responders, 39%) who underwent a new examination with contrast medium was also relevant.

Interestingly, 7.1% (10 of 141) of responders with previous HR to ICM who underwent a new contrast-enhanced examination developed a new HR; the 10 responders who developed a new reaction to ICM were negative on skin tests; 9 out of 10 underwent premedication and 6 out of 10 received an iodinated contrast medium different from the one responsible for the previous HR.

A non-negligible number of new reactions (2 out of 24, 8.3%) were also observed in the group of responders with previous HR to GBCM, both occurring in patients who tested negative for skin tests. In addition, one of the two patients underwent premedication, and, in both cases, a GBCM different from the one responsible for the first reaction was injected.

## 4. Discussion

Our large study, with long follow-up, demonstrates (1) a predominance of mild to moderate HR to CM, particularly to iodinated CM, in the female population; (2) a significant prevalence of asthma, food allergy, non-allergic drug hypersensitivity and neurologic disease in patients with HRs to GBCM compared to the high evidence of cardiovascular disease and history of cancer in patients with HRs to ICM; (3) a marked delay in performing skin tests in patients with negative results; (4) iomeprol, iopamidol and gadobenic acid as culprit CM most involved in HRs in patients with positive skin tests; and (5) an interesting percentage of new HRs to CM in patients with negative skin tests, most of whom underwent premedication and infusion of a CM different from that responsible for the first HR.

Gender appears to be a crucial risk factor for hypersensitivity reactions to radiocontrast agents [18,19,20]. The first evidence of this correlation dates back to a work by Lasser et al. [21], who, in a large multi-institutional study, found a higher incidence of reactions in women aged 18 to 30 years. Subsequently, Lang et al. [22] documented an increased risk of ICM reactions in women. Recently, an Italian study enrolled 407 patients from nine allergy centers to describe characteristics of patients experiencing HRs to ICM [23]. The authors confirmed a female predominance of patients developing HR to ICM. Furthermore, the multivariate analysis showed that female gender was a significant risk factor for ICM hypersensitivity reactions.

According to our knowledge, this is the first study comparing the frequencies of comorbid allergic and non-allergic conditions between patients with HRs to ICM versus patients with HRs to GBCM. Our enrollment of 350 patients from a single allergy center shows that non-allergic drug hypersensitivity and allergic rhino-conjunctivitis were the most frequently reported comorbid allergic/hypersensitivity conditions. These data confirm the previous evidence of the role of risk factors attributed to allergic rhinitis and asthma in the development of HR to both types of contrast media (ICM and GBCM) [4,23,24,25,26,27,28,29,30]. In 2019, Cha and coworkers found that drug allergy, other allergic diseases and a family history of ICM-related HRs were predictors of HR occurrence in 196,081 patients from seven participating institutions [24]. Previously, Kobayashi et al. identified asthma severity as a risk factor for acute hypersensitivity reactions to contrast agents [26]. Interestingly, we found a higher frequency of HRs to GBCM in patients with asthma, food allergy and non-allergic drug hypersensitivity compared to patients with HRs to ICM; this could be partially explained by the different stability of gadolinium-based contrast agents compared to new-generation ICM [31,32,33]. As expected [4,23], a history of cancer and heart/cardiovascular diseases were the most frequently reported comorbid non-allergic conditions in the group of patients with HRs to ICM. Not surprisingly, neurologic diseases were more frequent in the group of HRs to GBCM.

Skin tests are a crucial step in the diagnostic approach to patients with a previous hypersensitivity reaction to contrast media [3,4,5,34,35]. However, overall skin test positivity could be limited for both IHRs and NIHRs. In fact, a recent meta-analysis by Yoon et al. [36] emphasized that pooled per-patient positive rates of skin tests were 17% in patients with IHRs to ICM, reaching 52% when patients showed severe IHR. Among patients with NIHRs to ICM, the positive rate was 26%. Regarding HRs to GBCM, the clinical impact of skin tests in patients with a previous HR is controversial. Ideally, an ST would be indispensable for management purposes if it allowed the identification of an agent to which a patient would not react. Published studies [37,38,39] have reported the utility of skin tests with low cross-reactivity among GBCM tested and subsequent safe repeat administration of a negatively tested GBCM. The importance of performing skin tests in a specific time interval (6 weeks to 6 months) after the reaction to CM is strongly recommended [40]. Our results support the impact of time latency on skin test outcome [14,41]. In fact, in our cohort, the median delay was greater for patients with negative STs.

Our large study identified iomeprol, iopamidol and gadobenic acid as the most frequent culprit agents in patients with positive STs. These results enrich the heterogeneity of the statistics concerning the contrast media most frequently involved in HRs. In 2019, Cha et al. found that iobitridol and iomeprol were associated with higher rates of HRs, whereas iopromide and iohexol were associated with lower rates [24]. More recently, Meucci et al. found iomeprol and iopromide as the most commonly reported culprit ICM [41]. Furthermore, the authors reported a relevant rate of HRs due to unknown contrast agents [41]. Likewise, our experience confirms the impact of a real-life study design on missing information regarding the culprit contrast agent, as reported also in a French study [42]. This methodological aspect also weighs significantly on the identification of the paramagnetic contrast medium. However, our study confirms the lower rate of HR in GBCM [43] compared to ICM, a greater involvement of linear non-ionic GBCM (gadobenic acid) [25] and a low rate of severe anaphylactic reactions (0.009% of all HRs), in line with literature data [1].

To date, fatal anaphylaxis towards modern CM is considered rare. Nonetheless, iopamidol is one of the six drugs with over 20% of their reported anaphylactic reactions ending in death [44]. Furthermore, there are few case reports of fatal anaphylaxis towards modern GBCM [45,46,47,48]. In 1995, Jordan et al. described the first case of rapid onset of anaphylactic reaction leading to fatal cardiovascular arrest after injection of gadopentetate dimeglumine [46]. In 2015, Takahashi et al. reported the first autopsied case report of fatal anaphylaxis due to gadoteridol [47]. More recently, Prieto-García et al. described a case of fatal anaphylaxis due to gadobenate [48].

To the best of our knowledge, this is the first large study including 5-year follow-up data from both patients with HRs to ICM and patients with HRs to GBCM. In particular, the telephone survey allowed us to include almost 90% of the enrolled population in the analysis. Interestingly, 7.1% of the responders with previous HRs to ICM, all with negative skin tests, developed a new hypersensitivity reaction when subjected to a new examination with contrast medium, despite premedication in 90% of cases and infusion of a different iodinated contrast medium in 60%. Similarly, Meucci et al. [41] found a rate of new HRs after ICM re-exposure in 10.3% of telephone survey responders. Furthermore, in our study, a non-negligible number of new reactions (8.3%) were also observed in the group of responders with previous HR to GBCM. Finally, our data fuel the debate on the efficacy and usefulness of premedication and the choice of alternative contrast medium as a measure to reduce the risk of new HRs [49,50,51].

This study has several limitations. First of all, the retrospective nature of the study influences the timing of the allergological evaluation compared to the first HR, the identification of the culprit contrast medium, the outcome of the skin tests, the homogeneity of the follow-up period and the consequent number of any new reactions.

Finally, the heterogeneity of the premedication protocol does not allow an assessment of its impact on new HRs.

## 5. Conclusions

Contrast media are increasingly used worldwide, with millions of explorations being performed every day. The search for alternative safe CM is critical, given that many patients require CM-based procedures for diagnosis and disease monitoring. Therefore, we need a global approach to distinguish between the different types of reactions (i.e., non-allergic and true allergic hypersensitivity reactions) and to define a synergistic management protocol among the various actors involved (in particular, allergists, pediatricians, radiologists and general practitioners).

## Figures and Tables

**Figure 1 biomedicines-10-00759-f001:**
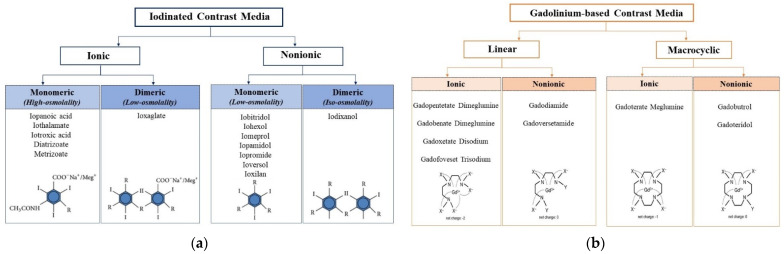
Classification of iodinated contrast media (**a**) and gadolinium-based contrast media (**b**) based on structural and chemical properties.

**Figure 2 biomedicines-10-00759-f002:**
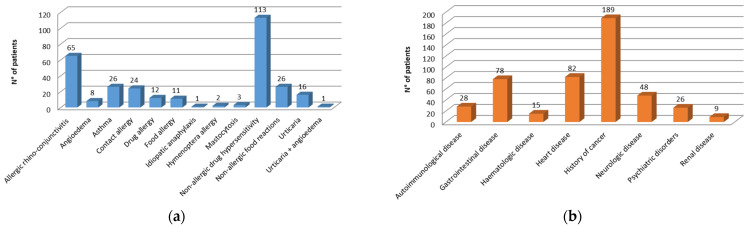
Frequency of comorbidities: (**a**) comorbid allergic/hypersensitivity; (**b**) comorbid non-allergic diseases.

**Table 1 biomedicines-10-00759-t001:** Demographic and clinical characteristics.

Enrolled Patients	(n°)	350
Female	(n°, %)	255 (73)
Age (years) at the first evaluation	(median, ±SD, range)	57 (±20, 7–89)
Delay (months) between reaction and ATs ^1^	(median, range)	62 (3–72)
Patients with HRs ^2^ to ICM ^3^	(n°, %)	278 (79)
Patients with HRs to GBCM ^4^	(n°, %)	65 (19)
Patients with HRs to unknown CM ^5^	(n°, %)	7 (2)
Total HRs	(n°)	368
IHRs ^6^	(n°, %)	294 (80)
Grade 1	(n°, %)	178 (60.5)
Grade 2	(n°, %)	67 (22.8)
Grade 3	(n°, %)	45 (15.3)
Grade 4	(n°, %)	3 (1)
Unknown	(n°, %)	1 (0.4)
NIHRs ^7^	(n°, %)	74 (20)
Mild	(n°, %)	9 (12.2)
Moderate	(n°, %)	65 (87.8)
Severe	(n°, %)	-

^1^: ATs = allergological tests; ^2^: HRs = hypersensitivity reactions; ^3^: ICM = iodinated contrast media; ^4^: GBCM = gadolinium-based contrast media; ^5^: CM = contrast media; ^6^: IHRs = immediate hypersensitivity reactions (see [13]); ^7^: NIHRs = non-immediate hypersensitivity reactions (see [14]).

**Table 2 biomedicines-10-00759-t002:** Comparison between patients with HRs to ICM and patients with HRs to GBCM.

		HRs ^1^ to ICM ^2^	HRs to GBCM ^3^	*p*-Value
Patients	(n°)	278	65	
Age (years) at the first evaluation	(median, ±SD)	59 (±16)	47 (±16)	<0.01 *
Total HRs	(n°, %)	289 (79) ^†^	72 (20) ^†^	N.S. *
IHRs	(n°, %)	226 (77) ^†^	61 (21) ^†^	N.S. *
NIHRs	(n°, %)	63 (85)	11 (15)	N.S. *
Comorbid allergic/hypersensitivity				**
Allergic rhino-conjunctivitis	(n°, %)	47 (16.7)	15 (21.7)	N.S.
Angioedema	(n°, %)	7 (2.5)	1 (1.4)	N.S.
Asthma	(n°, %)	15 (5.3)	10 (14.5)	<0.05
Contact allergy	(n°, %)	17 (6)	9 (13)	N.S.
Drug allergy	(n°, %)	9 (3.2)	2 (2.9)	N.S.
Food allergy	(n°, %)	5 (1.8)	6 (8.7)	<0.01
Idiopathic anaphylaxis	(n°, %)	1 (0.3)	-	-
Hymenoptera allergy	(n°, %)	1 (0.3)	1 (1.4)	N.S.
Mastocytosis	(n°, %)	3 (1.1)	-	-
Non-allergic drug hypersensitivity	(n°, %)	77 (27.4)	32 (46.4)	<0.01
Non-allergic food reactions	(n°, %)	19 (6.8)	5 (7.2)	N.S.
Urticaria	(n°, %)	10 (3.5)	6 (8.7)	N.S.
Urticaria + angioedema	(n°, %)	-	1 (1.4)	-
Comorbid non-allergic diseases				**
Autoimmunological disease	(n°, %)	21 (7.5)	6 (8.7)	N.S.
Cardiovascular/heart disease	(n°, %)	78 (27.7)	4 (5.8)	<0.01
Gastrointestinal disease	(n°, %)	64 (22.8)	13 (18.8)	N.S.
Hematologic disease	(n°, %)	10 (3.5)	3 (4.3)	N.S.
History of cancer	(n°, %)	159 (56.6)	28 (40.6)	<0.05
Neurologic disease	(n°, %)	31 (11)	16 (23.2)	<0.05
Psychiatric disorder	(n°, %)	21 (7.5)	5 (10.1)	N.S.
Renal disease	(n°, %)	7 (2.5)	1 (1.4)	N.S.

^1^: HRs = hypersensitivity reactions; ^2^: ICM = iodinated contrast media; ^3^: GBCM = gadolinium-based contrast media; *: Mann–Whitney U test; **: chi-square test; ^†^: The sum of percentages does not include reactions to unknown contrast media.

**Table 3 biomedicines-10-00759-t003:** Culprit contrast media.

HRs ^1^ to ICM ^2^	(n°)	289
iobitridol	(n°, %)	4 (1.4)
iodixanol	(n°, %)	5 (1.7)
ioversol	(n°, %)	1 (0.3)
iopamidol	(n°, %)	7 (2.4)
iomeprol	(n°, %)	52 (18)
iohexol	(n°, %)	8 (2.8)
iopromide	(n°, %)	50 (17.3)
unknown culprit ICM	(n°, %)	162 (56)
HRs to GBCM ^3^	(n°)	72
gadobutrol	(n°, %)	4 (5.5)
gadobenic acid	(n°, %)	6 (8.3)
gadoteridol	(n°, %)	7 (9.7)
gadopentetic acid	(n°, %)	1 (1.4)
gadoteric acid	(n°, %)	6 (8.3)
gadoxetic acid	(n°, %)	1 (1.4)
unknown culprit GBCM	(n°, %)	47 (65.3)

^1^: HRs = hypersensitivity reactions; ^2^: ICM = iodinated contrast media; ^3^: GBCM = gadolinium-based contrast media.

**Table 4 biomedicines-10-00759-t004:** Comparison between patients with HRs to ICM and patients with HRs to GBCM.

		Positive STs ^1^	Negative STs	*p*-Value
Patients	(n°)	30	320	
Time interval (months) between reaction and STs	(median, range)	22 (2–10)	66 (3–84)	<0.01 *
HRs ^2^ to ICM ^3^	(n°)	22	267	
IHRs	(n°, %)	19 (86.3)	207 (77.5)	N.S. **
NIHRs	(n°, %)	3 (13.6)	60 (22.5)	N.S. **
culprit ICM				**
iopromide	(n°, %)	6 (27.3)	44 (16.5)	N.S.
iomeprol	(n°, %)	10 (45.4)	42 (15.7)	<0.01
iopamidol	(n°, %)	3 (13.6)	4 (1.5)	<0.05
iohexol	(n°, %)	-	8 (3)	-
iobitridol	(n°, %)	1 (4.5)	3 (1.1)	N.S.
iodixanol	(n°, %)	-	5 (1.9)	-
ioversol	(n°, %)	-	1 (0.4)	-
unknown culprit ICM	(n°, %)	2 (9.1)	160 (59.9)	<0.01
HRs to GBCM ^4^	(n°)	8	64	
IHRs	(n°, %)	8 (100)	54 (84.4)	N.S. **
NIHRs	(n°, %)	0 (0)	60 (15.6)	N.S. **
culprit GBCM				**
gadobutrol	(n°, %)	1 (14.3)	3 (4.7)	N.S.
gadobenic acid	(n°, %)	3 (42.8)	3 (4.7)	<0.05
gadoteridol	(n°, %)	-	7 (10.9)	-
gadopentetic acid	(n°, %)	-	1 (1.6)	-
gadoteric acid	(n°, %)	1 (14.3)	5 (7.8)	N.S.
gadoxetic acid	(n°, %)	1 (14.3)	-	-
unknown culprit GBCM	(n°, %)	2 (25)	45 (70.3)	<0.01

^1^: STs = skin tests; ^2^: HRs = hypersensitivity reactions; ^3^: ICM = iodinated contrast media; ^4^: GBCM = gadolinium-based contrast media; *: Mann–Whitney U test; **: Fisher exact test.

**Table 5 biomedicines-10-00759-t005:** Follow-up: results from phone survey.

		ICM ^1^	GBCM ^2^
Responders	(n°)	246	61
New exams with CM ^3^	(n°, % of responders)	141 (57)	24 (39)
New HRs ^4^ to CM	(n°, % of pts who developed new exam)	10 (7.1)	2 (8.3)
Positive STs ^5^	(n°, % of pts who developed new HR)	0 (0)	0 (0)
Negative STs	(n°, % of pts who developed new HR)	10 (100)	2 (100)
Premedication	(n°, % of pts who developed new HR)	9 (90)	1 (50)
Same culprit CM ^6^	(n°, % of pts who developed new HR)	4 (40)	2 (100)

^1^: ICM = iodinated contrast media; ^2^: GBCM = gadolinium-based contrast media; ^3^: CM = contrast media; ^4^: HRs = hypersensitivity reactions; ^5^: STs = skin tests; ^6^: same contrast agent involved in the first HR.

## Data Availability

Not applicable.

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
