# Peer review of "Contrast Medium Hypersensitivity: A Large Italian Study with Long-Term Follow-Up"

_biomedicines, 2022, doi:10.3390/biomedicines10040759_

Round 1

Reviewer 1 Report

I have read with great interest the work "Contrast medium hypersensitivity: a large Italian study with 2 long-term follow-up".

The work has seemed very well planned, and although the primary objective is not very new, the secondary objective is.
The methods are well developed and explained and with well explained objectives.
The results are well explained with adequate tables and in which I only miss a figure of the chemical compositions that are spoken of to complete it.
The discussion is adequate and in the references I only miss mentioning some of the serious cases, even if they are anecdotal, one of them by the group of Dr Castells.

Author Response

March, 19th 2022

To Guest Editor and Reviewers
Biomedicines

We would like to greatly thank the Guest Editor and Reviewers who encouraged a revision of the manuscript.

Please find enclosed the Revision vers. 1 of the Original Article entitled “Contrast medium hypersensitivity: a large Italian study with long-term follow-up” by Eleonora Nucera, Giuseppe Parrinello, Sebastiano Gangemi, Alessandro Buonomo, Arianna Aruanno, Franziska Michaela Lohmeyer, Riccardo Inchingolo and Angela Rizzi.

Biomedicines-1646020 - Minor Revisions
Author's Reply to the Review Report (Reviewer 1)

Comments and Suggestions for Authors
I have read with great interest the work "Contrast medium hypersensitivity: a large Italian study with 2 long-term follow-up".
The work has seemed very well planned, and although the primary objective is not very new, the secondary objective is.
The methods are well developed and explained and with well explained objectives.
We thank the Reviewer for the comments.

The results are well explained with adequate tables and in which I only miss a figure of the chemical compositions that are spoken of to complete it.
We thank the Reviewer for the comments. We expanded the Introduction session by adding the definition of contrast media and the differences between the two classes. Furthermore, we added a figure illustrating chemical and structural properties.

The discussion is adequate and in the references I only miss mentioning some of the serious cases, even if they are anecdotal, one of them by the group of Dr Castells.
We thank the Reviewer for the comments. We modified the Discussion section accordingly.

With the best regards,
Eleonora Nucera, Giuseppe Parrinello, Sebastiano Gangemi, Alessandro Buonomo, Arianna Aruanno, Franziska Michaela Lohmeyer, Riccardo Inchingolo and Angela Rizzi

Corresponding Author:
Riccardo Inchingolo, MD, PhD
UOC Pneumologia, Fondazione Policlinico Universitario A. Gemelli IRCCS. Largo A. Gemelli, 8 – 00168 – Rome, Italy.

riccardo.inchingolo@policlinicogemelli.it

Corresponding Author will receive all editorial communications
The authors declare that the manuscript, or specified parts of it, have not been and will not be submitted elsewhere for publication.

Reviewer 2 Report

Dear Editor,

I was requested to review the manuscript with the following title:

"Contrast medium hypersensitivity: a large Italian study with long-term follow-up"

The manuscript has shed light on the Hypersensitivity reactions (HRs) to contrast media (CM), in which the authors compared differences of HRs to iodinated contrast media (ICM) versus gadolinium-based contrast media (GBCM) in 350 patients. Asthma, food allergy, non-allergic drug hypersensitivity and neurologic disease were significantly more frequent in patients with HRs to GBCM. CM components such as Iomeprol, iopamidol and gadobenic were associated with HRs in the patients with positive skin tests.

In general, the manuscript seems to be interesting and fits the scope of the journal.

One general comment, please include the definition of CM in the introduction and describe the difference between ICM and GBCM.

Author Response

March, 19th 2022

To Guest Editor and Reviewers
Biomedicines

We would like to greatly thank the Guest Editor and Reviewers who encouraged a revision of the manuscript.

Please find enclosed the Revision vers. 1 of the Original Article entitled “Contrast medium hypersensitivity: a large Italian study with long-term follow-up” by Eleonora Nucera, Giuseppe Parrinello, Sebastiano Gangemi, Alessandro Buonomo, Arianna Aruanno, Franziska Michaela Lohmeyer, Riccardo Inchingolo and Angela Rizzi.

Biomedicines-1646020 - Minor Revisions
Author's Reply to the Review Report (Reviewer 2)
Comments and Suggestions for Authors
Dear Editor,

I was requested to review the manuscript with the following title: "Contrast medium hypersensitivity: a large Italian study with long-term follow-up".
The manuscript has shed light on the Hypersensitivity reactions (HRs) to contrast media (CM), in which the authors compared differences of HRs to iodinated contrast media (ICM) versus gadolinium-based contrast media (GBCM) in 350 patients. Asthma, food allergy, non-allergic drug hypersensitivity and neurologic disease were significantly more frequent in patients with HRs to GBCM. CM components such as Iomeprol, iopamidol and gadobenic were associated with HRs in the patients with positive skin tests.
In general, the manuscript seems to be interesting and fits the scope of the journal.
One general comment, please include the definition of CM in the introduction and describe the difference between ICM and GBCM.
We thank the Reviewer for the comment. We expanded the Introduction session by adding the definition of contrast media and the differences between the two classes. Furthermore, we added a figure illustrating chemical and structural properties.

With the best regards,
Eleonora Nucera, Giuseppe Parrinello, Sebastiano Gangemi, Alessandro Buonomo, Arianna Aruanno, Franziska Michaela Lohmeyer, Riccardo Inchingolo and Angela Rizzi

Corresponding Author:
Riccardo Inchingolo, MD, PhD
UOC Pneumologia, Fondazione Policlinico Universitario A. Gemelli IRCCS. Largo A. Gemelli, 8 – 00168 – Rome, Italy.

riccardo.inchingolo@policlinicogemelli.it

Corresponding Author will receive all editorial communications
The authors declare that the manuscript, or specified parts of it, have not been and will not be submitted elsewhere for publication.